# Biting behaviour and infectivity of *Simulium damnosum* complex with Onchocerca parasite in Alabameta, Osun State, Southwestern, Nigeria

**Lateef O. Busari**[1]*, **Olusola Ojurongbe**[2], **Monsuru A. Adeleke**[1], **Olabanji A. Surakat**[1], **Akeem A. Akindele**[2]

**1** Department of Zoology, Osun State University, Osogbo, Osun State, Nigeria, **2** Department of Medical Microbiology and Parasitology, Ladoke Akintola University of Technology, Ogbomosho, Oyo State, Nigeria

* lateef.busari@pgc.uniosun.edu.ng

## Abstract

A longitudinal study was carried out to investigate species composition, seasonal abundance, parity and transmission potential of *Simulium damnosum* complex in Alabameta community in Osun State, Southwestern, Nigeria. Adult *Simulium damnosum* complex were collected along Owena River, Alabameta, by two dark complexioned vector collectors from 07:00hr to 18:00hr weekly using collecting tubes from November 2014 to April 2015. The flies were morphologically identified and dissected for the purpose of detecting Onchocerca parasite using dissecting microscope. The Monthly Biting Rate (MBR) of flies was determined using World Health Organization standard formula. A total of four hundred and forty flies were collected during the study period with all of them identified as forest species of *Simulium damnosum* complex. There was significant variation in monthly collection of the flies with the month of November having the highest number of flies (194) (44%) while the month of April recorded the lowest number of flies (31) (7%) (p<0.05). The morning biting peak (09hr - 11hr) (137) was higher than the evening biting peak (15hr -17hr) (64) (p<0.05) while nulliparous flies (294) (67%) were more abundant than the parous flies (146) (33%) (p<0.05). There was absence of infection (zero infectivity) of the flies (p<0.05). The zero infectivity in the flies may plausibly indicate the possibility of zero transmission of *Onchocerca* parasite in the community which if sustained over a period of time may signify the possibility of onchocerciasis elimination. Also, the presence of forest species of the flies reduces the risk of resident's intense exposure to blinding savannah strain of onchocerciasis.

## Introduction

Among the filarial nematodes, the public health significance of *Onchocerca volvulus* cannot be overemphasized being the causative organism of the dreadful and debilitating disease

**Data Availability Statement:** All relevant data are within the manuscript and its Supporting information files.

**Funding:** The author(s) have received no specific funding for this work.

**Competing interests:** The authors have declared that no competing interests exist.

onchocerciasis. Human onchocerciasis (river blindness) causes blindness and severe dermatitis in Africa and Latin America [1] and is the second leading cause of blindness [2]. Onchocerciasis is transmitted by members of *Simulium damnosum* complex through their bite while taking a blood meal [3]. Nine sibling species of *Simulium damnosum* complex have been taxonomically identified and documented in West Africa. The species include *Simulium sirbanum*, *S. damnosum sensu stricto*, *Simulium dieguerense*, *Simulium sanctipauli*, *Simulium soubrense*, *Simulium squamosum*, *Simulium yahense*, *Simulium leonense*, *Simulium konkorense* [4]. The first three species are known as Savannah flies which transmit Savannah strain of *O. volvulus* while the rest belong to the forest group and transmit the Forest strain of the parasite which causes more of skin diseases than blinding disease [5,6]. The congregation of adult worms and subsequent fertilization to produce microfilariae in the subcutaneous tissues has been known to elicit nodules [7].

Onchocerciasis is known to be endemic in many tropical countries and over 18 million people are infected worldwide and 120 million people are at the risk of the disease [8]. Onchocerciasis is most common in Africa while Nigeria probably has the highest burden of the disease [9]. Furthermore, Southwestern Nigeria (according to APOC rapid epidemiological mapping in 2008) was identified as one of the onchocerciasis endemic regions of the country. Earlier epidemiological studies by our team revealed high endemicity of onchocerciasis at Alabameta, albeit with low community microfilarial load [10]. The present study therefore aimed to provide information on species composition, seasonal abundance, parity and infectivity of black flies with a view to understanding its public health implication in and in planning effective control strategies in the study area and Osun State, Nigeria at large.

## Materials and methods

Ethical approval was obtained from the Osun State University Health Research Ethics Committee under the Ministry of Health, Osun State while oral consent was obtained from participants.

The Community is located in Ife South Local Government of Osun State, Southwest, Nigeria with an area of 730 kilometer square and a population of 135,338(2006 census). Owena River transverses the community and stretches towards Ondo State. The river usually produces rapids during the wet season and serves as a conducive breeding site for *S. damnosum*. However, during the dry season, the volume of water is usually drastically reduced to the extent that the rocks in the river are visible but does not become completely dry at all. Water from the river is used by residents to serve their domestic needs such as bathing and washing.

Alabameta lacks the necessary social amenities such as electricity, pipe borne water, hospitals, good roads, mobile phone and internet network and recreation centers expected in a human habitat. The major occupation of residents is farming.

An intensive quest was made around the community to identify the breeding sites of black fly in the selected area. The major breeding site identified was the Owena River. The black flies were collected at the designated collection point along the bank of the river as well as the small bridge over it using human landing collectors weekly in accordance with the standard protocol between November 2014 and April, 2015. However, field study was not conducted in February due to pandemonium in the area which led to the vandalization of lives and properties.

Dark complexioned adults were used for human landing catches to eliminate the influence of colour variation in adult catches. Two adult fly collectors were positioned at the collection site between 0700hr and 1800hr to catch the flies and work alternately. The human vector collectors were instructed to expose the lower part of their legs and collect flies moving around

the exposed leg to perch for blood meal using catching tubes. The numbers of flies caught hourly were recorded accordingly.

Each of the collected flies was identified morphologically for classification into forest and savannah flies. The flies were identified using morphological characters as proposed by Wilson *et al.*, (1993). The morphological characters include the colour of the wing tuft, colouration of the forecoxa and abdominal tergite [11–14].

The flies were dissected for parity and parasite infectivity using dissecting pins and dissecting microscope. The ovary of each fly was dissected to determine the parity and flies found to be parous was further dissected for onchocerciasis parasite. The head, thorax and abdomen of each parous fly was separately dissected in accordance with [15] and [13]. The parous rate, biting rate and transmission potential of the flies was determined in accordance with [4].

The biting rate of the black flies was determined from the Monthly Biting Rate (MBR):

$$\mathbf{MBR} = (\text{number of flies caught X number of days in the month})/\text{number of catching days}$$

[4].

The data from the study was subjected to T-test and Chi-square to determine the significant difference in dynamics and transmission potential of black flies during the period of the study. All analysis was performed using SPSS version 17.

## Results

A total of 440 black flies were caught during the study period (between November and April, 2015) with the month of November having the highest number of flies (194) while the month of April recorded the lowest number of flies (31) (Fig 1). There was a significant decline in the number of flies collected in each month (p = 0.041; p<0.05). The reflection of this was observed in the reduction in MBR in each month. November recorded the highest MBR (1940 bites per fly) while April had the lowest MBR (465 bites per fly) (Fig 2).

Furthermore, all the flies caught during the study period were forest flies (Fig 3).

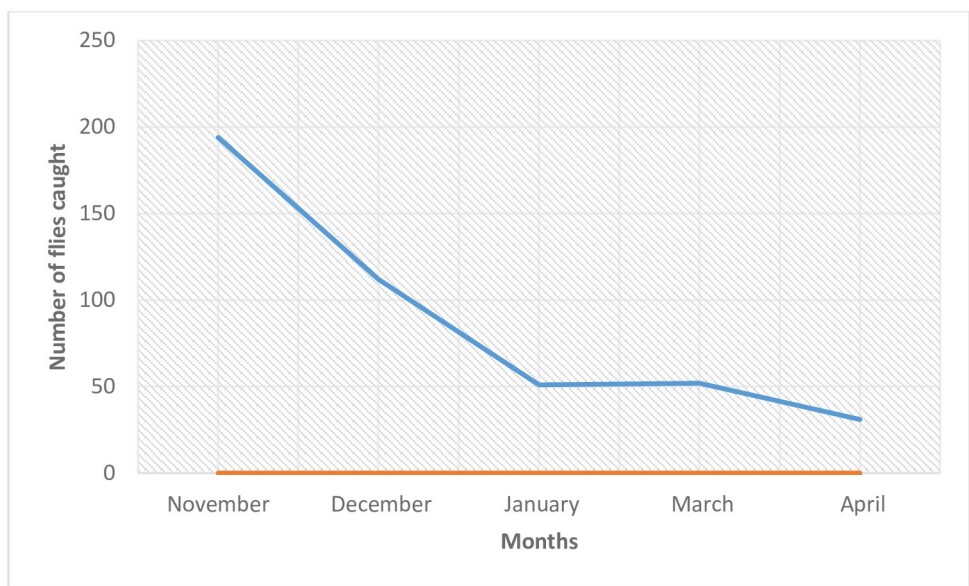

**Fig 1. Monthly distribution of *S. damnosum* in the study area.**

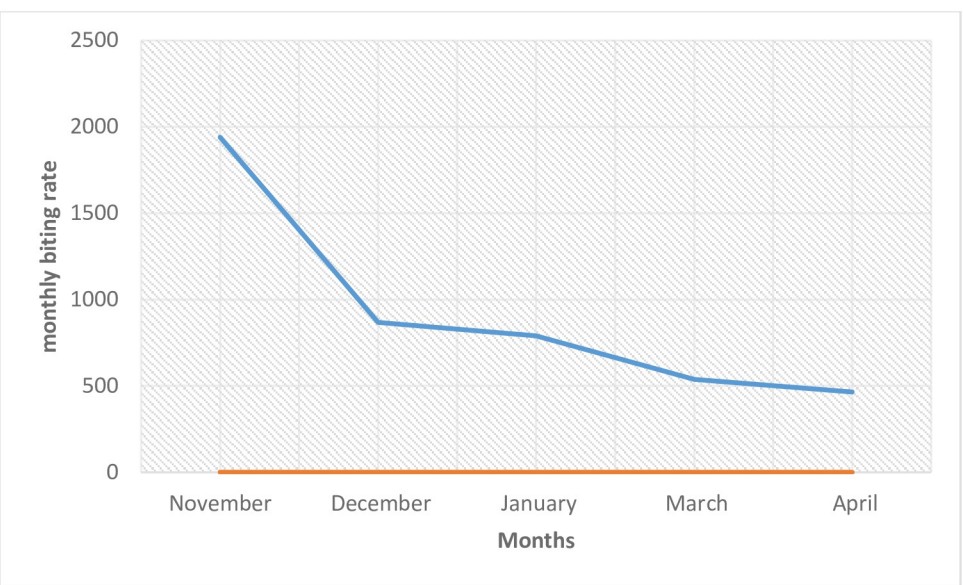

**Fig 2. Monthly biting rate of *Simulium damnosum* complex in the study area.**

There was a noticeable variation in the morphotaxonomic features of the adult flies. Most of the flies had completely pale wing tuft 01 or A (29%), 05 or E (27%), 02 or B (22%), 04 or D (19%) and 03 or C (3%).

The biting rhythms of the flies indicated that there was a significant variation in their biting behaviour on an hourly basis. Two biting peaks were observed at the study site. The biting activities of the flies were higher between 09hr– 11hr and 15hr–17hr in all the months during the study period except in November where the second biting peak was observed between 13hr– 15hr (Fig 4). Furthermore, the biting peak between 09hr– 11hr was significantly higher

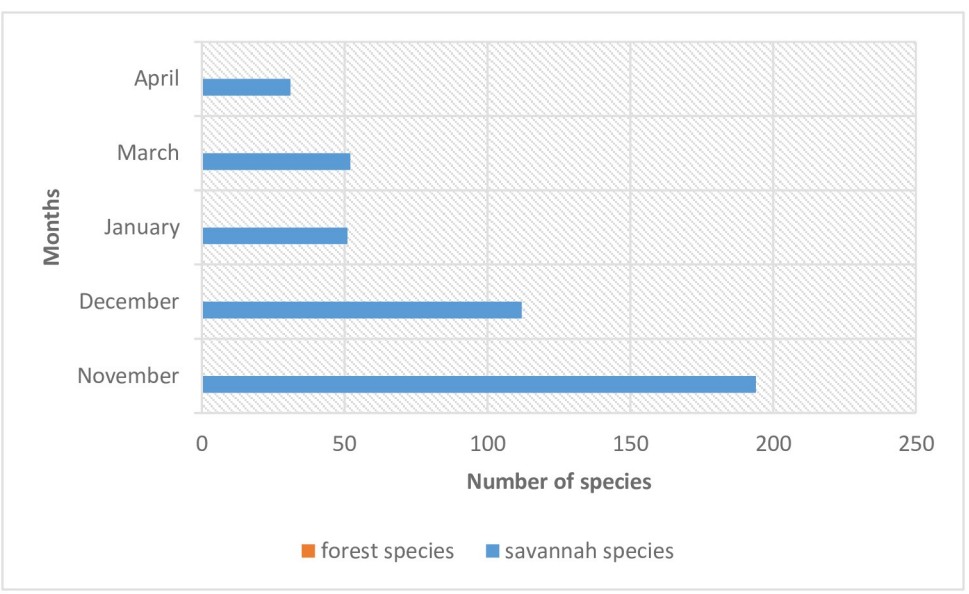

**Fig 3. Monthly distribution of species into forest and savannah in the study area.**

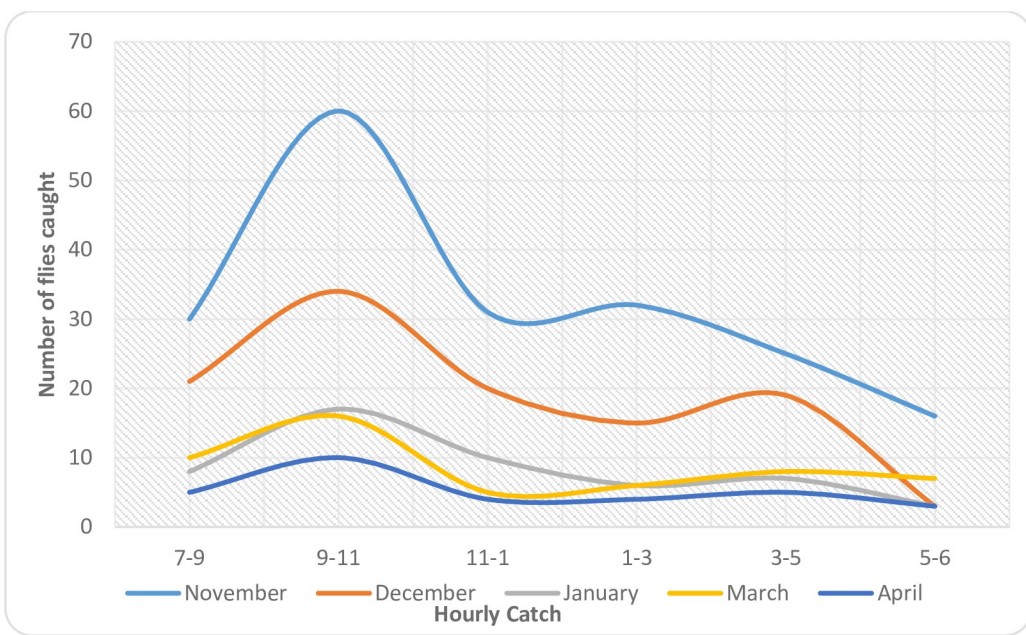

**Fig 4. Monthly biting rhythm of the flies in the study area.**

in November and January (p<0.05) as compared to other months. However, the evening biting peak was not statistically significant in all the months during the study period (p>0.05). The difference between morning and evening peaks were statistically significant throughout the period of the study (p<0.05).

From the 440 flies dissected, 146 (33.2%) were parous while 294 (66.8%) were nulliparous. None of the 146 parous flies was found to be infected with any stage of *O. volvulus* larvae during the study period (Table 1). November recorded the highest number of parous flies (84) (57.5%) and gradually declined subsequently till April (8) (5.5%) (Table 2). There was a significant difference in the number of parous and nulliparous flies collected throughout the period of the study (p = 0.004; p<0.05). The number of parous and nulliparous flies decreased on a monthly basis during the study period. The month of November recorded the highest number of both parous (84) and nulliparous (110) flies while April had the least (8 and 23 respectively).

## Discussion

Understanding the dynamics of insect vectors in an endemic area is important towards planning effective strategies to break vector-man The results of the present study showed that the

**Table 1. Transmission parameters of *Simulium damnosum* complex in the study area.**

| Variable | Transmission Parameters |
|---|---|
| Total flies dissected | 440 |
| No(%) of parous flies | 146(33.2) |
| No(%) of nulliparous flies | 294(66.8) |
| Month(%) with highest parous flies | November(57.5) |
| Month(%) with lowest parous flies | April(5.5) |
| No of flies infected with *O. volvulus* | 0 |
| Maximum Monthly Biting Rate | 1940(November) |
| Minimum Monthly Biting Rate | 465(April) |

**Table 2. Monthly variation in parous and nulliparous flies during the study period.**

| Month (%) | No of Parous flies (%) | No of Nulliparous flies (%) |
|---|---|---|
| November | 84(58) | 110(37) |
| December | 26(18) | 86(29) |
| January | 16(11) | 35(12) |
| March | 12(8) | 40(14) |
| April | 8(5) | 23(8) |

abundance of the flies usually coincide with the wet period, accompanied with high volume of the river and rapids. These conditions usually support the conducive breeding of black flies, thus leading to increase in adult flies population [16,17]. In contrast, the dry season is usually characterized with little or no rapids which could not support the breeding of the black flies. This factor, probably accounted for the low fly population recorded between January and April. This finding is consistent with previously published reports by [16–19], but contradicts the reports of [20] in Delta State and [21] in Owena Dam, Ondo State.

The results of morphotaxonomic studies showed that only the forest dwelling flies colonize and bite at Owena River, Alabameta Community during the study period. Alabameta community is located in a thick forest zone in Osun state. This could have accounted for the preponderance of forest dwelling flies in the area. However, the results is at variance with earlier results in Osun State, Southwestern Nigeria where the sympatric existence of forest and savannah dwelling flies have been reported [15,19,22].

The limitation of wing tuft as a sole diagnostic character was manifested in the variation of wing tuft in the forest flies at the study area. This could as well lead to a gross misidentification of forest flies as savannah flies based on pale wing tuft as earlier posited by [19]. It is however difficult to establish if the variations in wing tuft represent different cytospecies within forest dwelling species using morphotaxonomy. In West Africa, *S. damnosum s.l* is said to consist of nine sibling species [23]. Among the nine siblings, the distribution of *S. leonense*, *S. konkorense*, *S. dieguerense* and *S. sanctipauli* is restricted to only Sierra Leone, Guinea and Mali respectively [24,25]. Further observation on the flies showed that none of them had black trait of all the taxonomic characters, the presence of *S. yahense* could be excluded. *S. yahense* is known to be a dark fly [11]. It can therefore be inferred that the sibling species likely to be present in the Alabameta (along Owena River) are *S. squamosum* and *S. soubrense*.

Previous studies in Ogun, Oyan and Osun river systems in Southwestern Nigeria reported the preponderance of *S. soubrense* Beffa [15,22]. It is not unlikely that *S. soubrense* Beffa could also be the dominant forest flies at the study area. This however needs to be confirmed by molecular or cytotaxonomy through further studies.

Results from this study also showed that there was significant variation in the monthly biting activity of the flies at the study area. Moreover, bimodal peak was recorded for the daily biting rhythm of the flies. The first peak was between 09hr–11hr in all the months while the second was between 15hr–17hr except in November where it is between 13hr–15hr. Similar observations had earlier been reported in the literature [17,21,26,27]. However, the morning peak was higher than the evening peak. This observation dissented from the earlier reports where pronounced peak is usually recorded in the evening. The reason for this observation was not clear but not unlikely to be related to the behaviours of the people in the area. More people are active outdoor during the morning than the evening at the study area. This period coincides with peak of the biting of the flies. Apart from this, it could be as a result of environmental factors that were not monitored during this study. It has been observed that the environmental factors have great influence on the biting behaviours of the black flies [28]. The

influence of environmental factors on the biting behavior in the area could be a subject of further studies.

The results of dissection showed that majority of the flies caught at the study area during the study period were nulliparous. This observation is in consonance with the report of [20] but contrary to the report of [17,26,27]. The high proportion of the nulliparous flies may be as a result of the local production of the flies [17]. However, the decrease in the parous rate of the flies during the dry season could be attributed to the incomplete dryness of the breeding site (Owena River), thus supporting the development of flies and plausibly an indication of the absence of migratory flies. This observation is contrary to earlier reports by [16] and [15].

The absence of infection in the dissected flies may be an indication of low level transmission of onchocerciasis along Owena River. Among the factors that could have accounted for the low transmission are level of microfilariae in human hosts, number of surviving microfilariae in insects, vector-species complex and the survival rate of the flies [3,29–31].

Earlier epidemiological study at Alabameta showed very high prevalence of onchocerciasis but low microfilarial load [10]. [15] had reported that few out of hundreds of microfilariae ingested by the flies usually survive the effect of the peritrophic membrane that is secreted by the female flies after blood meals. This may be the reason that have accounted for zero infectivity recorded since the flies might not have ingested sufficient microfilariae to ensure continuous transmission.

This study has documented information on species composition and infectivity of *Simulium damnosum s.l* in Alabameta around Owena in Osun State. Further studies in adjoining communities around the river may be necessary to understand the dynamics and infectivity of onchocerciasis vectors in the area.

## Supporting information

**S1 File. Case study form.**
(DOCX)

## Acknowledgments

The authors appreciate and thank the fly collectors and field assistants for their support during the study. The Village Head and members of the community are all appreciated for their cooperation.

## Author Contributions

**Conceptualization:** Lateef O. Busari.

**Data curation:** Lateef O. Busari.

**Formal analysis:** Lateef O. Busari.

**Methodology:** Lateef O. Busari.

**Resources:** Akeem A. Akindele.

**Supervision:** Olusola Ojurongbe, Monsuru A. Adeleke.

**Writing – original draft:** Lateef O. Busari.

**Writing – review & editing:** Olabanji A. Surakat.

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
