## [Decision Letter · Decision Letter 0]

17 Mar 2021

PONE-D-20-39845

Biting behaviour and infectivity of Simulium damnosum complex with Onchocerca parasite in Alabameta, Osun State, Southwestern, Nigeria

PLOS ONE

Dear Dr. Busari,

Thank you for submitting your manuscript to PLOS ONE. After careful consideration, we feel that it has merit but does not fully meet PLOS ONE’s publication criteria as it currently stands. Therefore, we invite you to submit a revised version of the manuscript that addresses the points raised during the review process.

You are requested to address all issues raised by the expert reviewer.

Furthermore, you must provide better proof of the ethical review of the work, including submission of a copy of the ethical review approval, informed consent forms and case record forms (last two English Translations). These can be uploaded as suipplementary material.

In addition, you are requested to improve the quality of the figures.

We look forward to receiving your revised manuscript.

Kind regards,

Henk D. F. H. Schallig, Ph.D

Academic Editor

PLOS ONE

Journal Requirements:

Reviewers' comments:

Reviewer's Responses to Questions

**Comments to the Author**

1. Is the manuscript technically sound, and do the data support the conclusions?

Reviewer #1: Yes

2. Has the statistical analysis been performed appropriately and rigorously? 

Reviewer #1: Yes

3. Have the authors made all data underlying the findings in their manuscript fully available?

Reviewer #1: Yes

4. Is the manuscript presented in an intelligible fashion and written in standard English?

Reviewer #1: Yes

5. Review Comments to the Author

Reviewer #1: The manuscript reports on the biting behaviour and infectivity of Simulium damnosum complex with Onchocerca parasite in Alabameta, Osun State, Southwestern, Nigeria. As Nigeria is considered to have a very high burden of the disease globally, this study is both timely and relevant to onchocerciasis monitoring and control in the country.

The manuscript describes a technically sound piece of scientific research with data that supports the conclusions. Experiments were conducted rigorously following the established protocols, the statistical analysis is sound and the results were well discussed.

A few concerns though

Ethics: the authors state that approval was sought and obtained from the Oson Ministry of Health. They should please clarify whether a specific ethical clearance board at the Ministry gave the approval or it was just permission that was obtained from the ministry. It will also be ideal if the approval number is indicated.

Sample size and generalization: It is understood that this was a situational analysis of the biting behaviour and infectivity of Simulium damnosum complex with O. volvulus. However, the conclusions of zero infectivity should be made with caution as 6000 black flies may be needed to make a definitive conclusion on infectivity. A caveat can be made to this effect in the discussion.

Figures and Tables: The graphs should generally be improved. In Figure 2, the Y axis is labelled months, but the only label is ‘0’ and not months. Spacing between tables should be consistent.

6. PLOS authors have the option to publish the peer review history of their article (what does this mean?). If published, this will include your full peer review and any attached files.

Reviewer #1: No

---

## [Author Response · Author response to Decision Letter 0]

5 Apr 2021

1. The issue on Figure 2 has be revised. The months are now included in the y-axis, and consistency in the spacing within the figures.

2. The Ethical Clearance was obtained from UNIOSUN-HREC. It has been reflected in the manuscript and the letter is uploaded as supplementary document.

3. The statement on zero infectivity has been revised to reflect possibility of zero infectivity since our study did not deal with 6,000 flies.

---

## [Decision Letter · Decision Letter 1]

30 Apr 2021

PONE-D-20-39845R1

Biting behaviour and infectivity of Simulium damnosum complex with Onchocerca parasite in Alabameta, Osun State, Southwestern, Nigeria

PLOS ONE

Dear Dr. Busari,

Thank you for submitting your manuscript to PLOS ONE. After careful consideration, we feel that it has merit but does not fully meet PLOS ONE’s publication criteria as it currently stands. Therefore, we invite you to submit a revised version of the manuscript that addresses the points raised during the review process.

There is one correction that youare required to make:

The only concern is lines 52-54 and 65-67. The same statement on ethical clearance is made twice. Kindly correct this.

We look forward to receiving your revised manuscript.

Kind regards,

Henk D. F. H. Schallig, Ph.D

Academic Editor

PLOS ONE

Journal Requirements:

Reviewers' comments:

Reviewer's Responses to Questions

**Comments to the Author**

1. If the authors have adequately addressed your comments raised in a previous round of review and you feel that this manuscript is now acceptable for publication, you may indicate that here to bypass the “Comments to the Author” section, enter your conflict of interest statement in the “Confidential to Editor” section, and submit your "Accept" recommendation.

Reviewer #1: All comments have been addressed

2. Is the manuscript technically sound, and do the data support the conclusions?

Reviewer #1: Yes

3. Has the statistical analysis been performed appropriately and rigorously? 

Reviewer #1: Yes

4. Have the authors made all data underlying the findings in their manuscript fully available?

Reviewer #1: Yes

5. Is the manuscript presented in an intelligible fashion and written in standard English?

Reviewer #1: Yes

6. Review Comments to the Author

Reviewer #1: The only concern is lines 52-54 and 65-67. The same statement on ethical clearance is made twice. Kindly correct this.

7. PLOS authors have the option to publish the peer review history of their article (what does this mean?). If published, this will include your full peer review and any attached files.

Reviewer #1: No

---

## [Author Response · Author response to Decision Letter 1]

17 May 2021

The comment regarding the repetition of the ethical clearance statement twice have been rectified by removing the one previously on lines 65-67 and retaining the one preceding it.

---

## [Editor Report · Decision Letter 2]

20 May 2021

Biting behaviour and infectivity of Simulium damnosum complex with Onchocerca parasite in Alabameta, Osun State, Southwestern, Nigeria

PONE-D-20-39845R2

Dear Dr. Busari,

We’re pleased to inform you that your manuscript has been judged scientifically suitable for publication and will be formally accepted for publication once it meets all outstanding technical requirements.

Kind regards,

Henk D. F. H. Schallig, Ph.D

Academic Editor

PLOS ONE
---

## [Editor Report · Acceptance letter]

25 May 2021

PONE-D-20-39845R2 

Biting behaviour and infectivity of *Simulium damnosum* complex with Onchocerca parasite in Alabameta, Osun State, Southwestern, Nigeria 

Dear Dr. Busari:

I'm pleased to inform you that your manuscript has been deemed suitable for publication in PLOS ONE. Congratulations! Your manuscript is now with our production department. 

Kind regards, 

on behalf of

Dr. Henk D. F. H. Schallig 

Academic Editor

PLOS ONE